# [Re] Learning Fair Graph Representations via Automated Data Augmentations

## Abstract

We evaluate the reproducibility of the paper "Learning Fair Graph Representations via Automated Data Augmentations" by Ling et al. (2023). Our objective is to reproduce the three major claims that (1) fair augmentations improve fairness while retaining similar accuracy compared to other fairness methods, (2) augmenting both edges and node features performs better than augmenting only one of the two, and (3) learned augmentations reduce node-wise sensitive homophily and correlation between node features and the sensitive attribute. The authors provide an implementation of their method in PyTorch. We use and extend the given code, implementing an additional multi-run evaluation protocol with different random seeds. We further create additional baselines by disabling fairness in the model and investigating the generalizability of the method to other graph neural network (GNN) architectures and graphs with varying homophily. We partially reproduce claims (1), (2), and (3), attaining similar performance for two out of the three datasets originally used, as well as noisy results for the third dataset. Additionally, in our work, the correlation between node features and the sensitive attribute does not drop as significantly as in the original paper. On the other hand, we find that the method generalizes to other GNN structures yet does not generalize to graphs with varying homophily, failing for unbalanced homophily settings. Overall, the outcomes of the experiments indicate a lack of stability in the Graphair framework.

## 1 Introduction

Graph Neural Networks (GNNs; Kipf & Welling, 2017) have shown considerable performance (Gao & Ji, 2019; Liu et al., 2021a;b; Yuan et al., 2021) across multiple domains (Hamaguchi et al., 2017; Liu et al., 2022; Han et al., 2022; Hamilton et al., 2017). Despite this success, GNNs have been shown to carry over or amplify bias present in training data (Dai & Wang, 2021; Mehrabi et al., 2021; Olteanu et al., 2019). Specifically, the output of neural networks can be substantially correlated with sensitive features such as ethnicity, nationality, and gender, even though this behavior might not have been intended (Dai & Wang, 2021). Such biases can be present in graph data, hidden in both the way nodes are connected and their features (Mehrabi et al., 2021; Olteanu et al., 2019). Moreover, removing sensitive features does not always lead to a fairer dataset, as the sensitive information can be retrieved from other features or their combinations (Sweeney, 2002).

To tackle this issue, previous approaches for fair graph models have employed various pre-processing techniques (i.e. augmenting the data to remove bias before training the actual network). Some augmentation-based methods used handcrafted heuristics for changing the training data (Spinelli et al., 2021; Kose & Shen, 2022). Building upon this foundation, Ling et al. (2023) propose *Graphair*, a framework focused on learning data augmentations that remove bias from a dataset. The authors claim that this method considerably improves on common group fairness metrics across multiple datasets compared to other fair graph training schemes while achieving similar performance in terms of accuracy. We reproduce the experiments by Ling et al. (2023) to verify their main claims, as well as extend the tests of the method to examine its generalizability and robustness.

## 2 Scope of Reproducibility

Ling et al. (2023) propose a technique that automatically learns augmentations that change both the adjacency matrix and the node embeddings of a graph by incorporating adversarial and contrastive learning approaches. We identify the following claims from the paper:

**Claim 1:** Learned augmentations **improve group fairness metrics** while retaining **similar accuracy** compared to other fair graph-learning methods, on datasets containing sensitive attributes.

**Claim 2: Augmenting both edges and node features** achieves higher accuracy and better fairness scores than just augmenting one of them.

**Claim 3:** Learned augmentations **reduce node-wise sensitive homophily and correlation** between node features and the sensitive attribute.

## 3 Methodology

We use the code provided by the authors[1], with various improvements, such as setting random seeds for reproducibility, putting all the hyperparameters into a single file to make further replication and experimentation easier, adding missing functionality that was described in the paper, and bug fixes. It is important to mention that the code provided by the authors is not the original code they used for running the experiments. Therefore, we attempt to make the code as similar to the settings described in the paper as possible. We also extend the author's work with further experiments, which we describe in section 3.4. The code for the reproduction and extensions is available online[2].

### 3.1 Graphair framework overview

Given a graph $G = \{A, X, S\}$ with an adjacency matrix $A$, node features $X$, and the sensitive feature $S$, the Graphair method employs an augmentation module $g$ to produce fair augmented data, an adversary $k$ to ensure fairness, and a GNN-based encoder $f$ to produce node-embeddings for downstream tasks. The three components are described in detail below.

#### 3.1.1 Augmentation Module

The augmentation module $g$ first uses a Graph Convolutional Network (GCN; Kipf & Welling, 2017) encoder $g_{enc}$ to obtain node embeddings $Z = g_{enc}(A, X)$. The embedding $Z$ is used to create a new graph $G' = \{A', X', S\}$

$$Z_A = \mathrm{MLP}_A(Z), \quad \widetilde{A'} = \sigma\left(Z_A Z_A^T\right), \quad A'_{ij} \sim \mathrm{Bernoulli}\left(\widetilde{A'}_{ij}\right) \text{ for } i, j = 1, \cdots, n$$

$$Z_X = \mathrm{MLP}_X(Z), \quad \widetilde{M} = \sigma\left(Z_X\right), \quad M_{ij} \sim \mathrm{Bernoulli}\left(\widetilde{M}_{ij}\right) \text{ for } i, j = 1, \cdots, n$$

$$X' = M \odot X$$

where $\mathrm{MLP}_A$ and $\mathrm{MLP}_X$ are two Multi-Layer Perceptrons (MLPs), $\sigma$ is the sigmoid function, and $\odot$ is the Hadamard product. $\mathrm{MLP}_A$ and $\mathrm{MLP}_X$ both have two linear layers, separated by a ReLU activation function. To make the sampling process differentiable, the Gumble-Softmax parameterization trick is employed (Jang et al., 2016).

---

[1] https://github.com/divelab/DIG
[2] https://anonymous.4open.science/r/FACT-B421

### 3.1.2 Adversarial Training

To ensure fairness, Graphair uses adversarial training: a GCN with an MLP head is defined as the adversary $k : (A', X') \to \hat{S} \in [0, 1]^n$, which predicts the sensitive attribute $S$ from the augmented adjacency matrix $A'$ and masked node embeddings $X'$. The adversarial module $k$ is trained together with the augmentation module $g$ in an adversarial fashion, utilizing an adversarial loss $L_{\text{adv}}$. While the adversarial module tries to predict the sensitive feature from $A'$ and $X'$, the augmentation module is trained so that $A'$ and $X'$ don't contain bias, making it difficult for the adversary to recover the sensitive attribute. The adversarial training procedure can formally be described as the following optimization problem:

$$\min_g \max_k L_{\text{adv}} = \min_g \max_k -L_{\text{BCE}}\left(S_i, \hat{S}_i\right)$$

where $L_{\text{BCE}}$ denotes binary cross-entropy, $S_i$ denotes the sensitive feature and $\hat{S}_i$ denotes the prediction of the sensitive feature by the adversary.

### 3.1.3 Contrastive Learning and Reconstruction Loss

Additionally, to avoid the loss of information and converging to a trivial solution, a contrastive learning objective $L_{\text{con}}$ is used, which maximizes agreement between the representations of the original graph and the augmented one. These representations are denoted $h$ and $h'$ respectively and are learned by the GNN-based encoder $f$:

$$L_{\text{con}} = \frac{1}{2n} \sum_{i=1}^{n} [l\left(h_i, h'_i\right) + l\left(h'_i, h_i\right)]$$

where $l\left(a, b\right)$ denotes the contrastive learning objective defined by Zhu et al. (2020).

Finally, a reconstruction loss term $L_{\text{rec}}$ is added to avoid large deviations of the augmented graph from the original graph.

$$L_{\text{rec}} = L_{\text{BCE}}\left(A, \widetilde{A'}\right) + \lambda L_{\text{MSE}}\left(X, X'\right)$$

where $L_{\text{BCE}}$ and $L_{\text{MSE}}$ denote binary cross entropy and mean squared error (Goodfellow et al., 2016), $\widetilde{A'}$ refers to the edge probability matrix obtained from the augmentation module, and $\lambda$ is a hyperparameter.

The overall training process can be described by the following min-max optimization problem:

$$\min_{f,g} \max_k L = \min_{f,g} \max_k \alpha L_{\text{adv}} + \beta L_{\text{con}} + \gamma L_{\text{rec}}$$

where $\alpha$, $\beta$ and $\gamma$ are hyperparameters. This scheme defines a self-supervised training regime that aims to produce a fair GNN-based encoder $f$ as the end product for being used in downstream tasks. Appendix 5 shows a visual overview of the framework.

## 3.2 Evaluation

To evaluate the Graphair framework, the embeddings produced by the trained encoder are used for a downstream task. For this purpose, a simple MLP classifier is trained on the embeddings to predict a binary target attribute $Y$. The classifier's performance is measured by its test accuracy, as well as two group fairness metrics: Demographic Parity $\Delta\text{DP}$ and Equal Opportunity $\Delta\text{EO}$ (Louizos et al., 2015; Beutel et al., 2017), which are defined as follows:

$$\Delta\text{DP} = |P(\hat{Y} = 1|S = 0) - P(\hat{Y} = 1|S = 1)|$$

$$\Delta\text{EO} = |P(\hat{Y} = 1|S = 0, Y = 1) - P(\hat{Y} = 1|S = 1, Y = 1)|$$

where $\hat{Y}$ is the prediction of the classifier, $Y$ is the true target label, and $S$ is the sensitive feature. The MLP network consists of 2 linear layers with a ReLU activation in between. The classifier is trained 5 times and the average metrics are reported.

### 3.3 Datasets

For the replication, we use the same three datasets as the original paper: NBA, Pokec-z, and Pokec-n (Dai & Wang, 2020). We further create new synthetic datasets to verify the generalizability of the claims made by the authors.

#### 3.3.1 Original Paper Datasets

**NBA**: Nodes represent basketball players. Features include nationality, age, salary, and several performance statistics. The edges are based on whether players follow each other on social media. The sensitive feature is nationality, and the classification task is to predict whether a player's salary is above the median.

**Pokec**: The data comes from a popular social network in Slovakia, and is separated into two subsets by region: Pokec-z and Pokec-n. The nodes in the graph are the users, and the features are information about the users. The region of the users is treated as the sensitive feature. The classification task is to predict the working field of the users.

#### 3.3.2 Additional Datasets

**Synthetic Datasets**: To test Graphair's generalizability on graphs with varying homophily in a controlled way, we employ the DPAH model (Espín-Noboa et al., 2022), creating five different synthetic datasets. For each one, 1000 nodes are randomly sampled as 30% minority and 70% majority group, with group membership being the sensitive attribute. Additionally, each node has a predictive binary attribute income (high or low), and a binary attribute education (good or bad) that both correlate with the sensitive attribute. Specifically, a majority node has a 90% probability of having a high income while minority nodes have a 90% probability of having a low income. The majority nodes are 70% likely to have good education while minority nodes are 70% likely to have bad education. This defines an isolated setting where, even if the sensitive attribute were removed, the education value could be used to deduce the sensitive attribute and from there on deduce the income.

Hence, the network needs to mask both features to achieve full decorrelation. In total, 67% of nodes have a high income, so if full decorrelation is achieved, the accuracy should go to the same number. At the same time, $\Delta DP$ and $\Delta EO$ should go to zero because the conditional probability for the target feature given a sensitive feature does not change, as no information about sensitive features is available in this case. Five different variants are created to test graphs with different homophily. These vary in homophily among majority nodes and minority nodes.

Homophily refers to the tendency of similar nodes to connect (McPherson et al., 2001). Hence, the probability that two nodes $i$ and $j$ connect is determined by the homophily value between their classes $c_i$ and $c_j$, where a value lower than 0.5 means that $i$ is more likely to connect to a node from a different class than $c_j$, and a value higher than 0.5 means that $i$ is more likely to connect to a node of class $c_j$ than other classes. The specific definition of the corresponding probabilities is defined in the work by Espín-Noboa et al. (2022). Since we use two groups, i.e., majority $M$ and minority $m$, there are two homophily settings: $h_{MM}$ referring to the tendency of majority group nodes connecting, and $h_{mm}$ referring to how much minority groups tend to connect. Note that the between-group connectivity values are determined correspondingly: $h_{mM} = 1 - h_{mm}$ and $h_{Mm} = 1 - h_{MM}$.

Table 1 shows all relevant dataset statistics, as well as the specific homophily settings used for the synthetic dataset creation.

Table 1: Dataset statistics. Synthetic datasets are made up of 700 majority and 300 minority nodes. The first number in the synthetic dataset names refers to $h_{MM}$ while the second refers to $h_{mm}$.

| Dataset | Nodes | Node features | Edges | Inter-group edges | Intra-group edges | $h_{MM}$ | $h_{mm}$ |
|---------|-------|---------------|-------|-------------------|-------------------|----------|----------|
| **NBA** | 403 | 39 | 16,570 | 4,401 | 12,169 | | |
| **Pokec-z** | 67,797 | 59 | 882,765 | 39,804 | 842,961 | - | - |
| **Pokec-n** | 66,569 | 59 | 729,129 | 31,515 | 697,614 | | |
| **Syn-0.2-0.2** | | | | 23,695 | 6,275 | 0.2 | 0.2 |
| **Syn-0.2-0.8** | | | | 20,470 | 9,500 | 0.2 | 0.8 |
| **Syn-0.5-0.5** | 1,000 | 3 | 29,970 | 12,557 | 17,413 | 0.5 | 0.5 |
| **Syn-0.8-0.2** | | | | 9,107 | 20,863 | 0.8 | 0.2 |
| **Syn-0.8-0.8** | | | | 5,841 | 24,129 | 0.8 | 0.8 |

### 3.4 Experimental setup

#### 3.4.1 Original Experiment

**Claim 1**: Following the methodology of the authors, we conduct a grid search to find a hyperparameter setting of $\alpha$, $\gamma$, and $\lambda$ (over the values 0.1, 1, and 10) for the loss function that results in the best model in terms of average accuracy over 5 evaluation runs. The experiment is repeated for all three datasets: NBA, Pokec-z, and Pokec-n. Due to the size of Pokec datasets being significantly larger than others, batch training is used, where, following the method described in (Zeng et al., 2020), one batch is created using random walk sampling with 1000 root nodes and walk length 3.

**Claim 2**: The verification of this claim is addressed in an ablation study: we run our evaluation protocol while disabling the respective components of the augmentation module individually. Appendix 6 shows a graphic of this alteration.

**Claim 3**: To address this claim, we investigate how node-wise sensitive homophily (i.e., the tendency of nodes to connect with nodes having the same sensitive attribute value) and Spearman correlation between the sensitive and other features decrease in the fair setting.

#### 3.4.2 Change of Experimental Setup for Better Robustness Assessment

As described in Section 3.2, to evaluate the Graphair framework the authors train the models $f$, $k$, and $g$ once in an unsupervised manner, and then train a classifier 5 times to obtain the mean and variance of the results. However, this procedure does not show how robust and stable the actual proposed method (Graphair) is, but only how much variation in results the simple classifier undergoes throughout different runs. A more appropriate approach would be to test the stability of Graphair itself. Therefore, we conduct a second experiment, where models $f$, $k$, and $g$ are trained five times, together with the classifier. All further experiments use this approach. Diagrams of the two evaluation protocols are shown in appendix A.

#### 3.4.3 Further Experiments to Verify Claims

To further test whether the proposed method improves fairness in comparison to unfair methods (**Claim 1**), we carry out two additional experiments.

**Examining the effects of adversarial training**: The Graphair model is trained with hyperparameter $\alpha = 0$, which corresponds to not using the adversarial loss. Under such a setting, we expect fairness to deteriorate, and accuracy to improve (compared to training with a nonzero value like $\alpha = 1$) because the augmentation module no longer makes use of adversarial training to satisfy the fairness property. Appendix 7 shows a visual overview of this alteration.

Table 2: The results on the NBA, Pokec-z, and Pokec-n datasets, showing the original and the reproduced results with an identical experimental setup, and the results employing our proposed evaluation protocol.

| DATASET | SETTING | EVALUATION PROTOCOL | ACCURACY ↑ | Δ DP ↓ | Δ EO ↓ |
|---------|---------|---------------------|------------|--------|--------|
| NBA | original | original | $69.36 \pm 0.45$ | $2.56 \pm 0.41$ | $4.64 \pm 0.17$ |
| | reproduction | original | $\mathbf{69.64 \pm 0.69}$ | $\mathbf{0.14 \pm 0.00}$ | $3.98 \pm 0.87$ |
| | reproduction | ours | $69.36 \pm 0.46$ | $2.57 \pm 1.13$ | $\mathbf{2.33 \pm 1.53}$ |
| Pokec-z | original | original | $\mathbf{68.17 \pm 0.08}$ | $2.10 \pm 0.17$ | $\mathbf{2.76 \pm 0.19}$ |
| | reproduction | original | $61.83 \pm 0.29$ | $\mathbf{1.48 \pm 0.79}$ | $3.97 \pm 1.06$ |
| | reproduction | ours | $60.60 \pm 1.97$ | $2.84 \pm 2.04$ | $3.59 \pm 1.96$ |
| Pokec-n | original | original | $\mathbf{67.43 \pm 0.25}$ | $\mathbf{2.02 \pm 0.40}$ | $\mathbf{1.62 \pm 0.47}$ |
| | reproduction | original | $63.58 \pm 0.30$ | $3.80 \pm 0.43$ | $3.83 \pm 0.60$ |
| | reproduction | ours | $62.64 \pm 0.84$ | $3.82 \pm 2.99$ | $3.98 \pm 2.60$ |

**Comparing to supervised training**:

Graphair comprises numerous components, leading to a significant computational training expense. In order to assess the actual benefits of employing Graphair, we compare our initial results with those obtained through a fully supervised approach. Specifically, we bypass Graphair and instead train a GCN with the identical architecture as $f$, incorporating a classifier head on the original data. Appendix 8 shows a visual overview of this alteration.

### 3.4.4 Extensions to Verify Generalizability

To test the generalizability of the proposed method, the following additional experiments are conducted:

**Generalizability to different GNN architectures**: To investigate whether the method generalizes beyond GCNs to other Message-Passing GNNs (Gilmer et al., 2017), the experimental setup is modified, replacing all GCNs with Graph Attention Networks (GATs; Veličković et al., 2017). This verifies **Claim 1** in regards to whether the high accuracy and good fairness results hold in this case. Due to computational constraints, a full grid search is omitted at this point. Instead, the best hyperparameters from the original grid search are utilized.

**Generalizability to graphs with varying homophily**: We examine whether the method generalizes to different graph structures. To that end, the experiment is repeated on all five synthetic datasets described in Section 3.3.

### 3.5 Computational Requirements

We used a single NVIDIA RTX A6000 GPU for running the experiments. On this GPU, the average Graphair runtime per hyperparameter setting is around 3 minutes for the Pokec datasets and around 30 seconds for the NBA dataset. Running a full grid search for all three datasets combined takes around 14 GPU hours. All experiments combined took a total of 53.76 GPU hours.

Table 3: Setting alpha to zero, compared to its optimal value obtained through the grid search.

| DATASET | $\alpha$ | ACCURACY ↑ | Δ DP ↓ | Δ EO ↓ |
|---------|----------|------------|--------|--------|
| NBA | 0 | $69.08 \pm 1.71$ | $5.09 \pm 2.30$ | $5.26 \pm 2.54$ |
| | optimal | $\mathbf{69.36 \pm 0.45}$ | $\mathbf{2.57 \pm 1.13}$ | $\mathbf{2.33 \pm 1.53}$ |
| Pokec-z | 0 | $\mathbf{61.39 \pm 0.95}$ | $3.14 \pm 1.69$ | $4.97 \pm 1.13$ |
| | optimal | $60.60 \pm 1.97$ | $\mathbf{2.84 \pm 2.04}$ | $\mathbf{3.59 \pm 1.96}$ |
| Pokec-n | 0 | $60.65 \pm 0.62$ | $8.76 \pm 0.61$ | $8.85 \pm 0.84$ |
| | optimal | $\mathbf{62.64 \pm 0.84}$ | $\mathbf{3.82 \pm 2.99}$ | $\mathbf{3.98 \pm 2.60}$ |

Table 4: Comparison of the fully supervised method to Graphair.

| DATASET | SETTING | ACCURACY ↑ | Δ DP ↓ | Δ EO ↓ |
|---------|---------|------------|--------|--------|
| NBA | supervised | $68.65 \pm 2.03$ | $4.82 \pm 1.93$ | $2.93 \pm 2.41$ |
| | Graphair | $\mathbf{69.36 \pm 0.45}$ | $\mathbf{2.57 \pm 1.13}$ | $\mathbf{2.33 \pm 1.53}$ |
| Pokec-z | supervised | $\mathbf{70.21 \pm 0.49}$ | $8.93 \pm 1.41$ | $8.53 \pm 1.45$ |
| | Graphair | $60.60 \pm 1.97$ | $\mathbf{2.84 \pm 2.04}$ | $\mathbf{3.59 \pm 1.96}$ |
| Pokec-n | supervised | $\mathbf{68.51 \pm 0.29}$ | $\mathbf{1.37 \pm 0.79}$ | $\mathbf{1.27 \pm 0.95}$ |
| | Graphair | $62.64 \pm 0.84$ | $3.82 \pm 2.99$ | $3.98 \pm 2.60$ |

## 4 Results

### 4.1 Results Verifying Original Claims

**Claim 1 (group fairness improvement and similar accuracy)**: In table 2, we show the results of running grid search with the original paper's evaluation protocol and our proposed one, as described in section 3.2. On the NBA dataset, the results are fully reproduced with both evaluation protocols. Measurements even show a slight improvement on accuracy and fairness metrics in some cases. The accuracy on the Pokec datasets drops by a few percentage points (around 8% for Pokec-z and 5% for Pokec-n), while the fairness slightly worsens in most cases. We note, however, that these results are reasonably close to the original paper and the deviation can be attributed to the mini-batch training procedure. While the results show a balanced performance within the accuracy-fairness trade-off, we discuss best accuracy results can be found in appendix B and accuracy-fairness trade-off plots can be found in appendix C.

Table 3 shows the results of disabling adversarial training as described in section 3.4.3. We observe lower DP and EO values, which provide evidence for the reproducibility of **Claim 1**.

Table 4 shows the results of the fully-supervised setting, where we remove Graphair from the pipeline. Compared to Graphair, fairness is worse in the supervised setting on the NBA and Pokec-z datasets, but better on Pokec-n, providing partial evidence for **Claim 1**. Further discussion on these results can be found in section 5.

**Claim 2 (augmenting both edges and node features improves performance)**: Table 5 shows the results of the ablation study. For the NBA and Pokec-z datasets, **Claim 2** is reproducible, while for Pokec-n,

Table 5: Comparisons among different components in the augmentation model.

| DATASET | SETTING | ACCURACY ↑ | Δ DP ↓ | Δ EO ↓ |
|---|---|---|---|---|
| NBA | Graphair w/o FM | $68.79 \pm 1.19$ | $3.33 \pm 2.43$ | $2.82 \pm 2.31$ |
| | Graphair w/o EP | $\mathbf{71.06 \pm 3.09}$ | $6.32 \pm 1.46$ | $3.65 \pm 1.73$ |
| | Graphair | $69.36 \pm 0.45$ | $\mathbf{2.57 \pm 1.33}$ | $\mathbf{2.33 \pm 1.53}$ |
| Pokec-z | Graphair w/o FM | $60.21 \pm 0.95$ | $5.46 \pm 2.49$ | $6.07 \pm 1.88$ |
| | Graphair w/o EP | $\mathbf{69.32 \pm 0.31}$ | $9.87 \pm 2.25$ | $9.13 \pm 1.69$ |
| | Graphair | $60.60 \pm 1.97$ | $\mathbf{2.84 \pm 2.04}$ | $\mathbf{3.59 \pm 1.96}$ |
| Pokec-n | Graphair w/o FM | $60.60 \pm 1.59$ | $5.53 \pm 2.96$ | $5.21 \pm 2.29$ |
| | Graphair w/o EP | $\mathbf{67.51 \pm 0.20}$ | $\mathbf{1.24 \pm 1.00}$ | $\mathbf{0.92 \pm 1.31}$ |
| | Graphair | $62.64 \pm 0.84$ | $3.82 \pm 2.99$ | $3.98 \pm 2.60$ |

Table 6: Results for GAT network juxtaposed to the normal GCN setting.

| DATASET | SETTING | ACCURACY ↑ | Δ DP ↓ | Δ EO ↓ |
|---|---|---|---|---|
| NBA | GAT | $62.84 \pm 1.24$ | $9.16 \pm 8.97$ | $10.53 \pm 12.69$ |
| | GCN (normal) | $\mathbf{69.36 \pm 0.45}$ | $\mathbf{2.57 \pm 1.13}$ | $\mathbf{2.33 \pm 1.53}$ |
| Pokec-z | GAT | $60.42 \pm 2.82$ | $6.64 \pm 2.67$ | $6.21 \pm 2.37$ |
| | GCN (normal) | $\mathbf{60.60 \pm 1.97}$ | $\mathbf{2.84 \pm 2.04}$ | $\mathbf{3.59 \pm 1.96}$ |
| Pokec-n | GAT | $59.34 \pm 1.36$ | $4.47 \pm 1.12$ | $\mathbf{3.31 \pm 2.04}$ |
| | GCN (normal) | $\mathbf{62.64 \pm 0.84}$ | $\mathbf{3.82 \pm 2.99}$ | $3.98 \pm 2.60$ |

Graphair without edge perturbation achieves the best results. Possible reasons for this behavior are to be discussed in section 5.

**Claim 3 (Graphair reduces node-wise sensitive homophily and feature-sensitive correlation)**: Figures 1 and 2 show a recreation of figures 3 and 4 of the original paper. Figure 1 shows the same trend of decreasing node-wise sensitive homophily as in the original work. However, figure 2 exhibits far less decorrelation between the sensitive and the other features for the fair view of the original graph.

## 4.2 Results for Generalizing the Method

**Graph-Attention**: Table 6 shows the results of replacing all GCNs with GATs. Performance is consistently slightly worse for the GAT variants yet in a range of at most seven accuracy and eight fairness percentage points. Specifically, the method fails to achieve the same accuracy and fairness on the NBA dataset, whereas the performance is only slightly off for the Pokec-z and Pokec-n datasets.

**Synthetic Datasets**: The results for the five different synthetic datasets can be seen in table 7. It shows high accuracy and low fairness for the dataset variants with homophily values of 0.2 and 0.2, as well as 0.8 and 0.8, respectively. The first setting implies that within-group connections are scarce while between-group connections are ample, whereas the latter setting implies ample within-group and low between-group connections. On the other hand, the other homophily settings achieve an accuracy equal to the prior of 0.67, as 67% of nodes exhibit high income. Simultaneously, fairness is optimal in these cases at 0.00 $\Delta DP$

Table 7: Results for different artificial datasets.

| DATASET | ACCURACY ↑ | Δ DP ↓ | Δ EO ↓ |
|---|---|---|---|
| Syn-0.2-0.2 | $70.60 \pm 1.74$ | $23.47 \pm 10.72$ | $19.09 \pm 10.52$ |
| Syn-0.2-0.8
Syn-0.5-0.5
Syn-0.8-0.2 | $67.00 \pm 0.00$ | $0.00 \pm 0.00$ | $0.00 \pm 0.00$ |
| Syn-0.8-0.8 | $77.20 \pm 5.88$ | $50.93 \pm 29.14$ | $54.55 \pm 31.49$ |

and $\Delta EO$ for Syn-0.2-0.8, Syn-0.5-0.5, and Syn-0.8-0.2. The homophily in these settings could be defined as more harmonized, and the method achieves full decorrelation of the output and the sensitive feature by also masking the third feature, which in turn is correlated with the sensitive feature.

## 5   Discussion

**Reproducibility**: Although the original paper does not test the entire method in a multi-run setting with different random seeds, **Claim 1** and **Claim 2** are largely reproduced using both the original and our proposed evaluation setup. However, both claims could not be fully reproduced for all datasets. In our experiments, Graphair does not produce better results on the Pokec-n dataset as compared to baselines that do not account for fairness. The ablation study is also not fully reproduced, as for the Pokec-n dataset, Graphair without edge perturbation produced better results than the full method. In addition to that, **Claim 3** could only be partially reproduced, as node-wise sensitive homophily was shown to decrease, but Spearman correlation between the sensitive feature and other features did not significantly decrease in our experiments. We conclude that **Claim 1**, **Claim 2** and **Claim 3** are partially reproducible. We note that throughout multiple experiments, we observe that the method is not stable, which could hurt its adoption for a variety of tasks.

**Generalizability**: We conduct several additional experiments to verify the generalizability of the method. Firstly, we show that other message-passing GNN architectures perform reasonably well on the two larger datasets (Pokec-z and Pokec-n) while the results are considerably worse for the NBA dataset. These results are in range, as we use the same hyperparameters that were optimal for the normal Graphair method. Performing another hyperparameter grid search for the GAT variant could lead to better performance. We conclude that the method generalizes to other message-passing architectures and thus exhibits robustness regarding the graph architecture used.

Secondly, we show that the method does not generalize to graphs with varying homophily. Our five synthetic datasets exhibit different levels of connectivity within groups and in between groups. On one hand, three of our datasets have harmonized connectivity, i.e., if intra-group connections are scarce, inter-group connections are plenty, or both are balanced. On the other hand, there are two extreme variations where there are extremely scarce connections within and in between groups, or many intra- and inter-connections. Specifically, the method can achieve full decorrelation for harmonized homophily values for the majority and minority groups, while extreme values (very low inter-group or intra-group connectivity) lead to failure. This shows that not every dataset could be used in conjunction with Graphair to achieve fairness, as harmonized homophily characteristics can leave substantial bias in the dataset even after augmentation through Graphair.

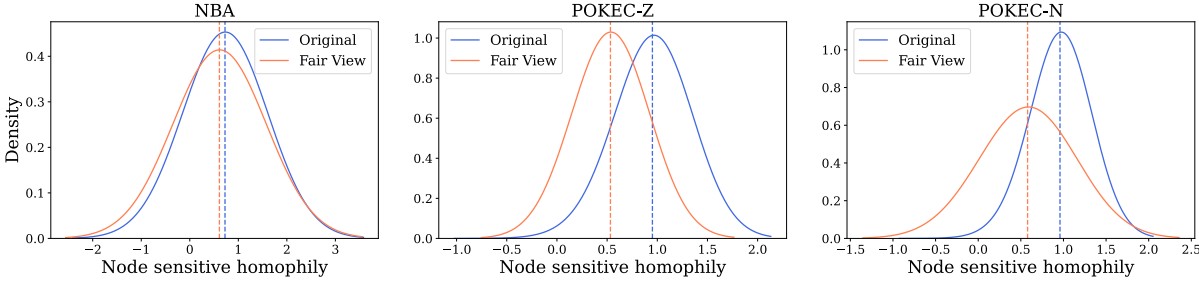

Figure 1: Distributions of node-wise sensitive homophily in the original and the fair graph data.

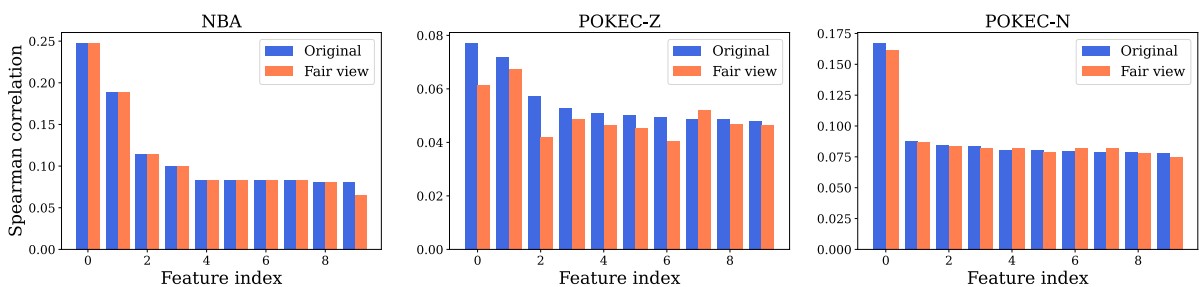

Figure 2: Spearman correlation between the sensitive feature and the other node features in the original and the fair graph data.

### 5.1 What was easy

The study design was mostly straightforward, with very clear and well-written explanations in the paper. The availability and ease of use of the code also contributed to this. That way, we were able to run the experiments described in the original paper without major problems. In addition, the claims were clearly presented in the original paper, aiding our work.

### 5.2 What was difficult

The code had several shortcomings that posed problems to a full reproduction. Firstly, random seeds were only set for the dataset shuffling procedures and evaluation, whereas the training of the networks in the Graphair method did not have the random seed set. Secondly, there were various hardcoded hyperparameter settings in the code, many of which were not mentioned in the paper. Lastly, some parts of the experimental code were challenging to execute, such as the mini-batching procedure which required extensive work to set up and run. Also, several bugs were identified in the code, some of which were fixed by the authors.

### 5.3 Communication with original authors

We sent multiple emails to the authors with clarification questions, to which the authors swiftly and kindly answered each time and with clear explanations. The authors moreover improved their code base after our emails, supporting our work.

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

# A  Evaluation Protocol

.

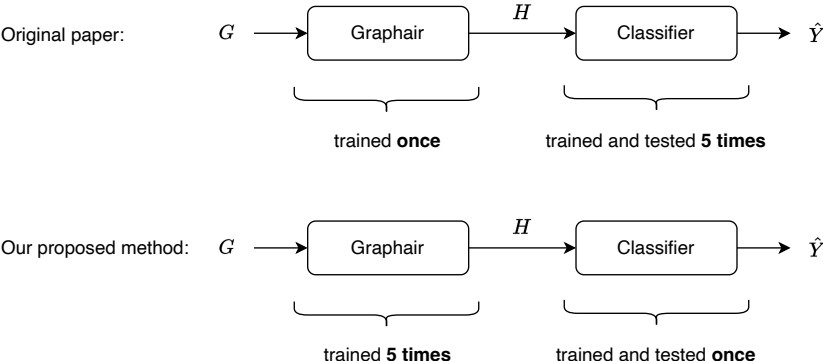

Figure 3: Evaluation protocol of the original paper and our proposed method.

# B  Grid Search Best Accuracy Results

Table 8: Best-accuracy results on the NBA, Pokec-z, and Pokec-n datasets, showing the original ones, the reproduced results with an identical experimental setup, and the results employing our proposed evaluation protocol.

| DATASET | SETTING | ACCURACY ↑ | Δ DP ↓ | Δ EO ↓ |
|---|---|---|---|---|
| NBA | original (original evaluation protocol) | $69.36 \pm 0.45$ | $\mathbf{2.56 \pm 0.41}$ | $4.64 \pm 0.17$ |
| | reproduction (original evaluation protocol) | $\mathbf{70.92 \pm 1.19}$ | $18.50 \pm 5.32$ | $9.30 \pm 3.63$ |
| | reproduction (our evaluation protocol) | $70.35 \pm 0.53$ | $4.75 \pm 2.60$ | $\mathbf{4.55 \pm 1.95}$ |
| Pokec-z | original (original evaluation protocol) | $\mathbf{68.17 \pm 0.08}$ | $\mathbf{2.10 \pm 0.17}$ | $\mathbf{2.76 \pm 0.19}$ |
| | reproduction (original evaluation protocol) | $62.01 \pm 0.03$ | $3.74 \pm 0.69$ | $4.93 \pm 1.01$ |
| | reproduction (our evaluation protocol) | $60.94 \pm 2.34$ | $3.73 \pm 1.43$ | $4.36 \pm 2.68$ |
| Pokec-n | original (original evaluation protocol) | $\mathbf{67.43 \pm 0.25}$ | $\mathbf{2.02 \pm 0.40}$ | $\mathbf{1.62 \pm 0.47}$ |
| | reproduction (original evaluation protocol) | $63.59 \pm 0.30$ | $3.80 \pm 0.43$ | $3.83 \pm 0.60$ |
| | reproduction (our evaluation protocol) | $62.79 \pm 1.41$ | $5.48 \pm 2.62$ | $4.73 \pm 1.87$ |

## C Pareto-Fronts of Grid Searches

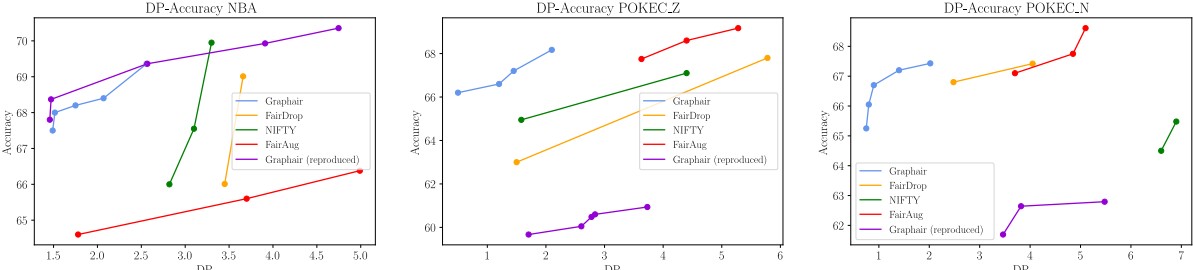

Figure 4: DP-Accuracy Pareto front plots for all three datasets. Graphair (reproduced) refers to the model trained with our proposed evaluation protocol.

## D Original Framework

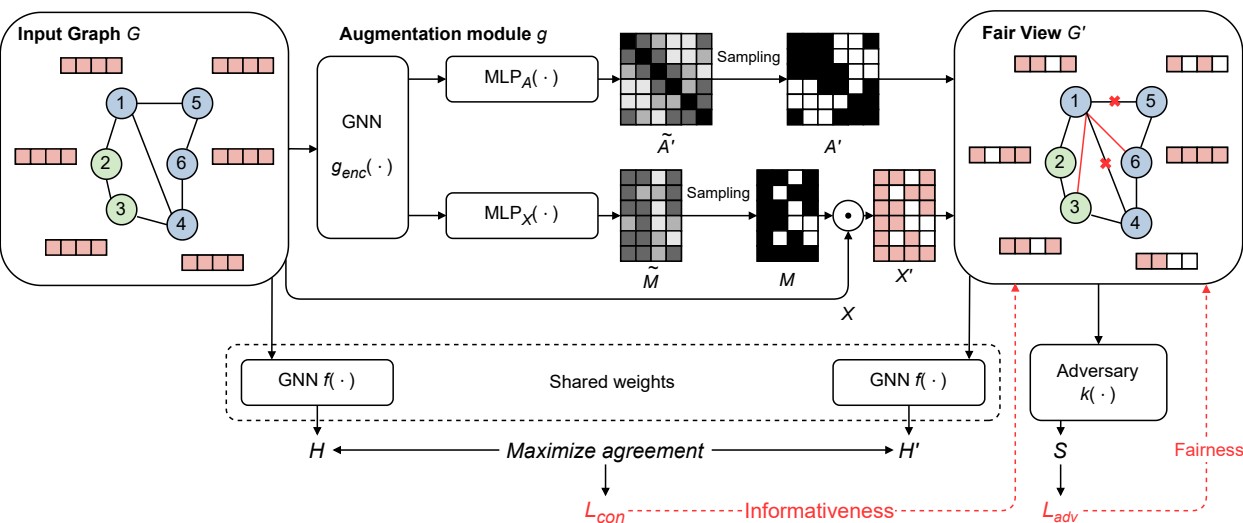

Figure 5: Graphair framework. The graphic was adapted from the original paper (Ling et al., 2023).

## E  Framework for Ablation Study

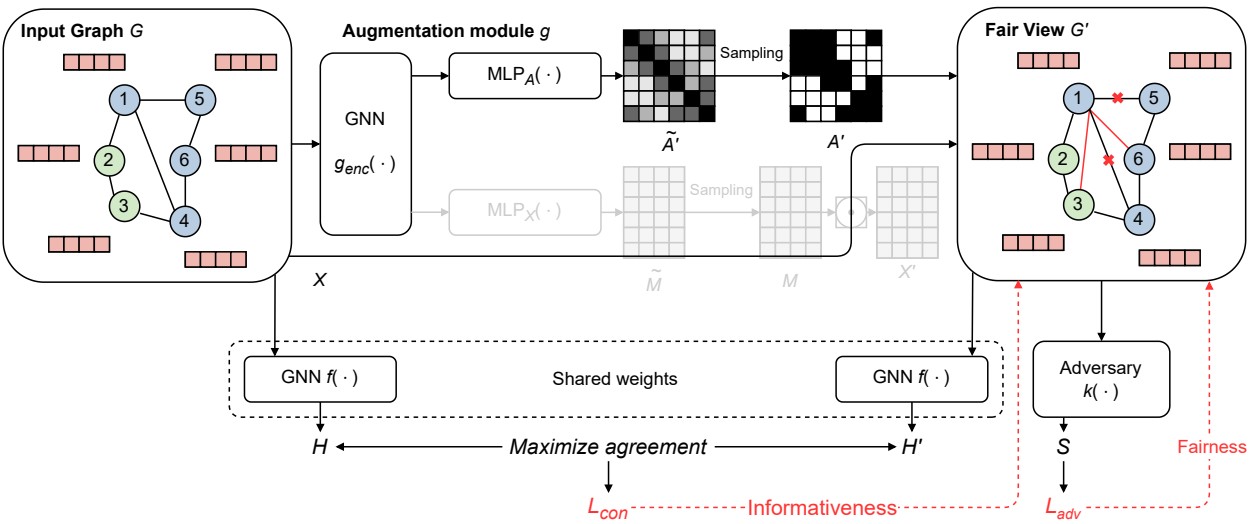

Figure 6: Graphair framework without node feature masking for the ablation study.

## F  Framework without Adversarial Training

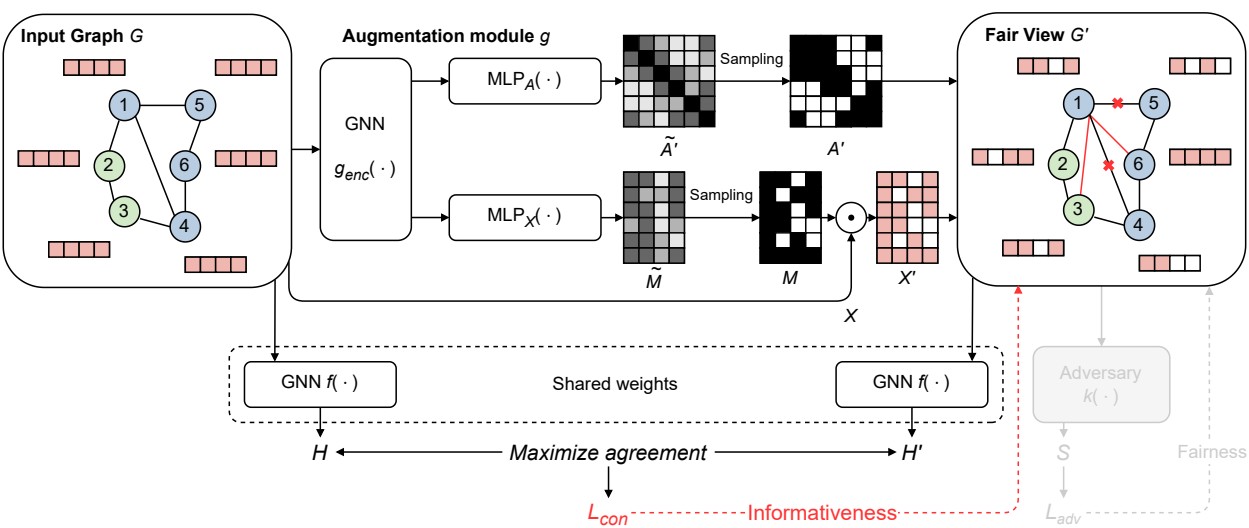

Figure 7: Graphair framework without the adversarial training.

# G   Framework for Supervised Training

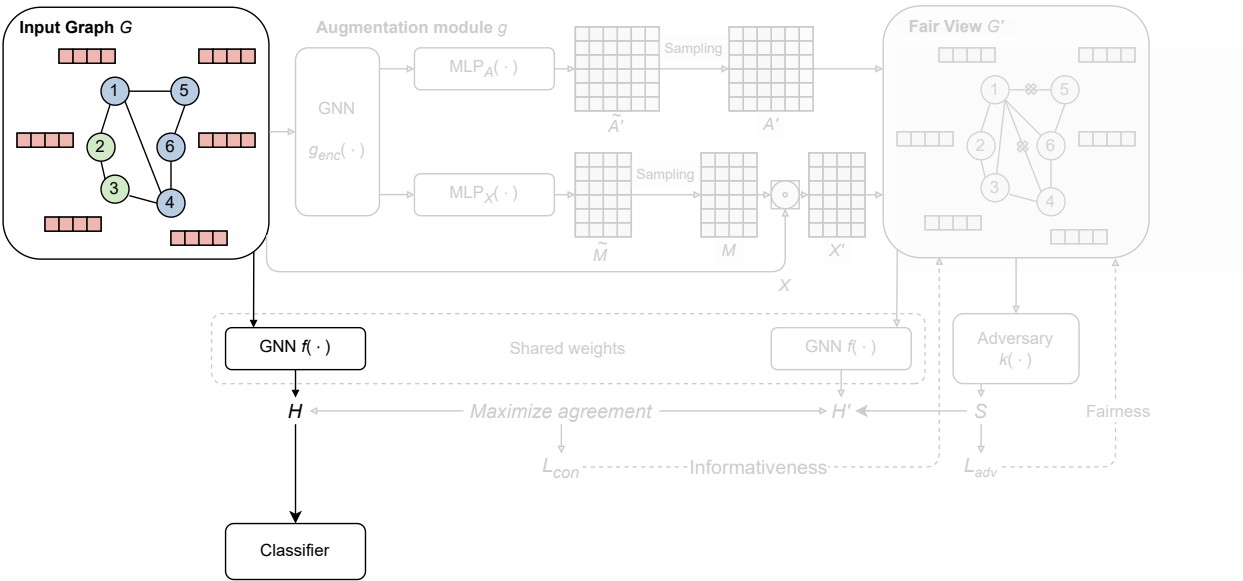

Figure 8: Fully-supervised training without Graphair framework.

