# OpenReview forum: "[Re] Learning Fair Graph Representations via Automated Data Augmentations"
_TMLR — Rejected by TMLR_

### Review · Reviewer_zesW · 2024-04-26

**Summary Of Contributions:**

The paper in question evaluates the reproducibility of “Learning Fair Graph Representations via Automated Data Augmentations” by Ling et al. (2023), focusing on three major claims related to fairness and accuracy in graph neural networks (GNNs). The reproducibility study extends the original PyTorch implementation with a multi-run evaluation protocol and additional baselines. The findings indicate that while some claims were partially reproduced, there were inconsistencies across datasets and a noted lack of stability in the Graphair framework.

**Audience:**

No

**Claims And Evidence:**

Yes

**Requested Changes:**

Please kindly refer to the Weaknesses.

**Strengths And Weaknesses:**

Strengths:

* The reproducibility study enhances the original work by implementing a multi-run evaluation protocol, providing a more robust assessment of the claims.
* The study explores the generalizability of Graphair to other GNN architectures and varying homophily levels, contributing to a broader understanding of Graphair’s applicability.
* Despite challenges, the study successfully reproduces partial of the original claims, affirming the potential of fair augmentations to improve fairness without compromising accuracy.


Weaknesses:

* My concerns are mainly about whether this manuscript is out of the scope of TMLR. The aim of this work focuses on reproducing the claims of Graphair. Although the results are interesting, this type of research does not fit the requirements of TMLR in my opinion. Other venues such as the ICLR blog track would be better for this paper.

* The contributions regarding research perspective are unclear, given the authors neither propose new approaches nor develop new benchmarks in addition to their observations from the reproducibility of Graphair. This makes this paper more like a technical report of experiments rather than a research paper.

* While the reimplementation of Graphair supports the authors’ observations that only partial of the claims in Graphair can be reproduced, it would be better for the authors to propose standard experimental setups as well as benchmarks based on these observations, which could then benefit the research in fair graph learning.

---

> ### Author Response · Authors · 2024-06-06
>
> Dear Reviewer,
>
> We greatly appreciate your feedback and suggestions.
>
> Your main concern was about our paper being "*inside the scope of TMLR*", as it is a reproducibility study. We conducted and submitted our study for the Machine Learning Reproducibility Challenge (MLRC; https://reproml.org/). The MLRC expects submissions that revolve merely around reproducing a former study, as well as assessing its generalizability. MLRC officially partners with TMLR (https://reproml.org/blog/announcing_mlrc2023/#on-the-road-ahead), meaning that all submissions have to be done at TMLR. Hence, we believe our work is in this year’s scope of TMLR. Moreover, we believe there is a reproducibility crisis in the field of machine learning. There is a significant amount of research published in reputable academic venues, such as TMLR, that remains largely unverified. Our primary contribution is adding to the body of reproducibility papers that aim to validate cutting-edge research, thereby ensuring its quality. In this specific case, our study uncovered numerous problems with the original code base and led the authors to fix critical bugs in their code, which is part of an open-source package for graph deep learning. This example highlights the general importance of reproducibility research. As TMLR officially partners with the MLRC, TMLR evidently finds reproducibility research important and sees assessing a study’s reproducibility and generalizability as a valuable contribution to ML research.
>
> Since the last revision, we performed major additions to our work based on your and the other reviewers’ feedback, detailed below. We reran most of our experiments, some of which at a larger scale. Accordingly, we updated the entire report. We also include the list of our main contributions:
>
> 1. We repeat all experiments from the original paper by Ling et al. (2023).
> 2. We alter the evaluation procedure to retrain the entire method five times with different random
> seeds, thereby achieving a more robust assessment of the method’s stability and performance.
> 3. We assess how well the method generalizes to different architectures.
> 4. We conduct further ablation studies to investigate how the individual components of the Graphair
> framework contribute to improving fairness.
> 5. We run additional experiments with synthetic datasets to assess the stability and performance of
> Graphair for datasets with different homophily.
> 6. We improve the readability and reproducibility of the given code.
>
> Unfortunately, OpenReview did not let us add a revised manuscript version. We think this functionality may have accidentally been disabled, and we have contacted the Action Editor for re-enabling it. As soon as it is re-enabled, we will update our manuscript in OpenReview. Until then, we host the updated report on PDF Host, which can be found at  https://pdfhost.io/v/7Snq1A4HC_TMLR_Journal_Submission
>
> We also add a changelog of all the adjustments we made:
>
> Abstract:
> - Stated contributions
> - Strengthened conclusions about instability in Graphair
>
> 1 Introduction:
> - Stated contributions
>
> 3.3.2 Additional Datasets
> - Rewrote entire section, providing more detailed explanations
>
> 3.4.3 Longer Training Procedure
> - Performed further experiments on a larger scale
> - Explained the longer training procedure
>
> 3.4.4 Further Experiments to Verify Claims
> - Changed explanation of ablation on adversarial training
> - Fully rewrote explanation about the comparison to unfair baseline
>
> 3.4.5 Additional Experiments to Verify Graphair’s Generalizability
> - Performed additional full grid search for GATs
> - Mentioned grid search in the experiment description
>
> 3.5 Computational Requirements
> - Updated GPU hours
>
> 4.1 Results Verifying Original Claims
> - Rewrote section with new results and conclusions
>
> Results Table 2
> - Added results for longer training runs
>
> Results Table 3
> - Reran experiments with longer training procedure and updated results
> - Adjusted table for better readability
>
> Results Table 4
> - Reran experiments with longer training procedure and updated results
>
> Results Table 6
> - Reran experiments with longer training procedure and updated results
>
> 4.2 Results Testing the Generalizability of the Graphair Method
> - Rewrote section with new results and conclusions
>
> 5 Discussion
> - Rewrote section with adjusted conclusions
> - Added sub-section discussing instability
>
> Appendix
> - Added section H for further information on homophily
> - Added section I detailing further experiments and results on larger batch sizes
>
> We hope these additions and explanations help clarify why we believe there is a clear audience for our work, and moreover alleviate the concerns voiced in your feedback.
>
> We thank you for your constructive suggestions.
>
> Kind regards

---

### Review · Reviewer_E1RF · 2024-04-29

**Summary Of Contributions:**

The paper studies the reproducibility of the paper “Learning Fair Graph Representations via Automated Data Augmentations”. The authors first study the fair augmentations of Graphair. Furthermore, the authors investigate the generalizability of the method to another graph neural network (GNN) architecture and graphs with varying homophily.

**Audience:**

No

**Broader Impact Concerns:**

I don't think there are any concerns about the ethical implications.

**Claims And Evidence:**

No

**Requested Changes:**

Please refer to the "Strengths And Weaknesses" section.

**Strengths And Weaknesses:**

S1. The proposed evaluation protocol can help reduce the variation of adversarial training and get more robust evaluation results.

S2. The authors conduct an ablation study on adversarial loss. This experiment can study the effects of adversarial training, which is the key to how Graphair achieves fairness. The experimental results show that adversarial debiasing is crucial for Graphair to achieve fairness.

W1. The comparison between supervised training and self-supervised representation learning, such as Graphair, is problematic. Supervised training leverages labels to guide the learning process, typically resulting in models that classify or predict based on explicit patterns associated with these labels. Conversely, self-supervised methods learn representations based on the inherent patterns and structures within the data, thus they can inadvertently amplify existing societal biases. For instance, while supervised learning might cluster data based on labels such as 'safe' or 'unsafe' areas in a city, self-supervised methods might cluster based on sensitive attributes such as neighborhood income levels or racial demographics. This kind of clustering can exacerbate biases and potentially degrade the fairness performance of self-supervised methods when compared to supervised learning. Therefore, drawing direct comparisons between supervised training and self-supervised representation learning can be inherently problematic.

W2. There seem to be issues in the mini-batch training code. On the NBA dataset, which does not require batch training, the reproduction results surpass those reported in the original paper. A potential issue with the provided code involves the training steps. When mini-batch training is used, the total number of training steps should be calculated as the number of training epochs multiplied by the steps per epoch, approximately $500 \times (66,000 / 3,000) = 11,000$ steps. However, the reproduction code only utilizes 500 steps. This discrepancy could cause Graphair to underfit, potentially leading to misleading results.

W3. In the study of the generalization capabilities of various GNN architectures, the authors applied the best hyperparameters determined from an initial grid search to other architectures. However, since adversarial training is known to be particularly sensitive to hyperparameter settings, the appropriateness of these parameters can vary significantly across different architectures. This variance could lead to results that might not convincingly demonstrate the generalization performance of other architectures.

W4. In the creation of new synthetic datasets, it's unclear why the probabilities of connectivity between groups differ, denoted as $h_{mM}\neq h_{Mm}$. Moreover, while homophily describes the tendency for similar nodes to connect, this is not observed in most new synthetic datasets. Instead, the phenomenon appears as anti-homophily, where nodes within the same group are less likely to connect. These scenarios are atypical in real-world settings, casting doubts on the utility of such configurations.

---

> ### Author Response · Authors · 2024-06-06
>
> Dear Reviewer,
>
> We are deeply grateful for your detailed and insightful suggestions. Based on your feedback, we have repeated and extended most of our experiments and performed a major overhaul of the paper. We detail our additions below, addressing your concerns alongside:
>
> Firstly, you were concerned about the” *comparison between supervised and self-supervised training*”, arguing that these both learn fundamentally different representations and thus cannot be compared. We thank you for pointing this out, as we believe we failed to properly communicate this experiment. Graphair consists of two components: fairness training to produce a new graph by disentangling sensitive attributes from all other features, and a subsequent supervised training part using a GNN classifier. Contrary to our faulty explanation, our experiment did not involve a comparison between self-supervised and supervised training, but rather involved the removal of the fairness training component in Graphair, instead training a classifier directly on the “unfair” graph. Please note that when Graphair is used, the evaluation is still performed by training a supervised classifier, the only difference is in the input graph, which is augmented by the Graphair framework. If no fairness training is involved, the method does not optimize for fairness and hence does not decorrelate sensitive attributes. Therefore, this experiment served as a sanity check for Graphair, revealing whether the fairness training actually had a notable effect on fairness metrics, which is the central claim in the original paper.
> Accordingly, we updated section 3.4.4, and would invite you to read the updated version.
>
> Secondly, you correctly pointed out to us that Graphair is not trained for sufficiently many steps on the Pokec datasets. Even though we used the code provided by the authors of the original paper, and thus the same hyperparameters, we additionally reran the experiments based on your feedback with 11,000 steps, detailed in the added section 3.4.3, as well as Table 2. We further performed a second additional scaling experiment, utilizing a maximally large batch size based on our available GPU’s memory, detailed in Appendix I. Based on these experiments, the results changed locally yet our conclusions remain the same.
>
> Thirdly, you rightly noted that we did not perform a full grid search on our GAT experiment. We executed a full grid search and updated sections 3.4.5, 4.2, as well as results Table 6 accordingly. The results for models with GAT improved, however our conclusions remain unchanged.
>
> In addition, you voiced concerns over our synthetic dataset experiments. Our explanations of these experiments fail to properly describe these experiments and, we believe, leave the reader utterly confused. Therefore, we have completely rewritten section 3.3.2. To summarize, we (1) created an isolated setting in which Graphair has to fully mask the entire input, and (2) varied the connectivity of the dataset, corresponding to, e.g., segregated minority subcultures that appear in many societies. We generally wanted to assess whether Graphair generalizes to varying types of graph datasets. We kindly invite you to read the rewritten section 3.3.2.
>
> Lastly, we want to address your concern about the *audience*.  We conducted and submitted our research for the Machine Learning Reproducibility Challenge (MLRC; https://reproml.org/). The MLRC expects submissions that revolve merely around reproducing a former study, as well as assessing its generalizability. MLRC officially partners with TMLR, meaning that all submissions will have to be done at TMLR. Hence, we believe our work is in this year’s scope of TMLR and has an audience.
>
> Unfortunately, OpenReview did not let us add a revised manuscript version. We think this functionality may have accidentally been disabled, and we have contacted the Action Editor for re-enabling it. As soon as it is re-enabled, we will update our manuscript in OpenReview. Until then, we host the updated report on PDF Host, which can be found at  https://pdfhost.io/v/7Snq1A4HC_TMLR_Journal_Submission.
>
> We made adjustments to the following sections: we changed the abstract, as well as sections 1, 3.3.2, 3.4.3, 3.4.4, 3.4.5, 3.5, 4.1, 4.2 and 5, as well as Tables 2, 3, 4, 6. We also changed appendix H and I.
>
> We hope these additions and explanations help clarify why we believe there is a clear audience for our work, and moreover alleviate the concerns voiced in your feedback.
>
> We greatly appreciate your detailed and thoughtful feedback and suggestions.
>
> Kind regards

---

### Review · Reviewer_JMHV · 2024-05-24

**Summary Of Contributions:**

The objective of this paper is to reproduce the Graphair algorithm proposed in the previous article, "Learning Fair Graph Representations via Automated Data Augmentations," and to evaluate the three main claims presented in that article. This paper partially reproduces the three claims from the previous article, achieving similar results on two out of three datasets. Finally, the authors point out that the Graphair algorithm in the original article exhibits instability issues.

**Audience:**

Yes

**Claims And Evidence:**

No

**Requested Changes:**

Weaknesses

**Strengths And Weaknesses:**

Strength
1.	The authors generated 5 additional synthetic datasets to evaluate the generalizability of Graphair algorithm.
2.	The authors modified the evaluation method from the original algorithm by increasing the training iterations for the augmentation module 𝑔, adversary 𝑘 and encoder 𝑓 from 1 to 5, thereby achieving a more robust assessment.
3.	The authors provided a detailed introduction to the framework of the original Graphair algorithm, which helps readers to understand it.

Weakness
1.	This paper revolves around the validation and supplementary experiments of an algorithm from a previously published article. While this may help researchers better understand the Graphair algorithm, it does not contribute to the field of fair graph learning, since it does not propose any new methods or perspectives.
2.	The authors did not conduct tests on additional fairness measures, such as Disparate Impact or Equalized Odds.
3.	The runtime of the Graphair algorithm on all three datasets was at most 3 minutes, indicating that the datasets used in the experiments were too small to adequately assess the scalability of the algorithm.

---

> ### Author Response · Authors · 2024-06-06
>
> Dear Reviewer,
>
> We want to thank you for your feedback. We detail our responses below:
>
> Firstly, you voiced your concern that our work may not be novel enough, being a reproducibility study. We conducted and submitted our study for the Machine Learning Reproducibility Challenge (MLRC; https://reproml.org/). The MLRC expects submissions that revolve merely around reproducing a former study, as well as assessing its generalizability. MLRC officially partners with TMLR (https://reproml.org/blog/announcing_mlrc2023/#on-the-road-ahead), meaning that all submissions will have to be done at TMLR. Hence, we believe our work is in this year’s scope of TMLR. Moreover, We believe there is a reproducibility crisis in the field of machine learning. There is a significant amount of research published in reputable academic venues, such as TMLR, that remains largely unverified. Our primary contribution is adding to the body of reproducibility papers that aim to validate cutting-edge research, thereby ensuring its quality. In this specific case, our study uncovered numerous problems with the original code base and led the authors to fix critical bugs in their code, which is part of an open-source package for graph deep learning. This example highlights the general importance of reproducibility research. As TMLR officially partners with the MLRC, TMLR evidently finds reproducibility research important and sees assessing a study’s reproducibility and generalizability as a valuable contribution to ML research.
>
> However, we nonetheless failed to communicate our contributions well. Hence, we updated the paper. We invite you to read the updated version. We also add a clear listing of our contributions:
>
> 1. We repeat all experiments from the original paper by Ling et al. (2023).
> 2. We alter the evaluation procedure to retrain the entire method five times with different random
> seeds, thereby achieving a more robust assessment of the method’s stability and performance.
> 3. We assess how well the method generalizes to different architectures.
> 4. We conduct further ablation studies to investigate how the individual components of the Graphair
> framework contribute to improving fairness.
> 5. We run additional experiments with synthetic datasets to assess the stability and performance of
> Graphair for datasets with different homophily.
> 6. We improve the readability and reproducibility of the given code
>
> Secondly, you noted that we did not test Graphair on additional fairness metrics, specifically Disparate Impact or Equalized Odds. Equalized Odds is a close variant of Equal Opportunities (https://fairlearn.org/main/user_guide/assessment/common_fairness_metrics.html#equal-opportunity), and Disparate Impact is a synonym for Demographic Parity (https://fairlearn.org/main/user_guide/assessment/common_fairness_metrics.html#demographic-parity). We assessed Graphair on Equal Opportunities as well as Demographic Parity. Moreover, our results show that individual fairness metrics did not make a notable difference as we found Graphair to be generally very unstable. Based on the feedback from all the reviewers, we have rerun all the experiments for a more robust assessment. We updated the report accordingly, especially sections 4, and 5, to better point out this instability in Graphair.
>
> Lastly, you correctly pointed out that we do not test Graphair at a large enough scale. Based on this feedback, we reran most of our experiments, but most importantly used (1) much larger batch sizes as well as (2) significantly larger numbers of steps (500 vs 11,000). The updated experiments are detailed in the added section 3.4.3, as well as results in Table 2, and Appendix I. We further performed an additional full grid search on the GAT variant to enlarge the scale of that experiment as well, detailed in section 3.4.5.
>
> Unfortunately, OpenReview did not let us add a revised manuscript version. We think this functionality may have accidentally been disabled, and we have contacted the Action Editor for re-enabling it. As soon as it is re-enabled, we will update our manuscript in OpenReview. Until then, we host the updated report on PDF Host, which can be found at  https://pdfhost.io/v/7Snq1A4HC_TMLR_Journal_Submission
>
> We made adjustments to the following sections: we changed the abstract, as well as sections 1, 3.3.2, 3.4.3, 3.4.4, 3.4.5, 3.5, 4.1, 4.2 and 5, as well as Tables 2, 3, 4, 6. We also changed appendix H and I.
>
> We hope these additional experiments as well as explanations help clarify why we believe there is a clear audience as well as sufficient contributions in our work. We moreover hope our additions can alleviate your further concerns.
>
> We thank you for your insightful feedback.
>
> Kind regards

---

### Decision · Action_Editor_uy7J · 2024-06-14

**Recommendation:** Reject

**Comment:**

Upon review, the conclusion does not significantly deviate from the original paper [1]. The reviewers and I believe that the extension of the experiments lacks clear insights into the original methodology. The paper focuses solely on validating and supplementing [1] without proposing new methods or perspectives. Additionally, its contribution as a benchmark paper is very limited.

The reviewers made several suggestions, such as testing additional fairness measures like Disparate Impact or Equalized Odds, or proposing more challenging benchmarks. However, the authors' response did not change the reviewers' opinions on these points.

Furthermore, one of the reviewers raised several concerns regarding some of the additional settings and experiments conducted in the paper. The authors' response did not address these concerns adequately.

In summary, this paper serves as a comprehensive technical report on [1]. However, the reviewers and I believe that the paper does not introduce a distinct contribution nor present new empirical findings that would be of interest to the research community.

[1] Ling et al. (2023)

**Audience:**

The paper has a very limited audience because the additional contributions beyond the original work are too narrow.

**Claims And Evidence:**

The paper replicates and expands upon the experiments conducted in "Learning Fair Graph Representations via Automated Data Augmentations" by Ling et al. (2023). More precisely, it presents an evaluation of the Graphair algorithm by generating additional synthetic datasets to assess its generalizability. The authors also improved the evaluation method by increasing the training iterations compare to the original paper. Furthermore, a detailed introduction to the original Graphair framework is provided.